# Biomarkers of the L-Arginine/Dimethylarginine/Nitric Oxide Pathway in People with Chronic Airflow Obstruction and Obstructive Sleep Apnoea

**DOI:** 10.3390/jcm12165230

**Published:** 2023-08-11

**Authors:** Juliane Hannemann, Elin H. Thorarinnsdottir, André F. S. Amaral, Edzard Schwedhelm, Lena Schmidt-Hutten, Heike Stang, Bryndis Benediktsdottir, Ingibjörg Gunnarsdóttir, Thórarinn Gislason, Rainer Böger

**Affiliations:** 1Institute of Clinical Pharmacology and Toxicology, University Medical Center Hamburg-Eppendorf, 20246 Hamburg, Germany; ju.hannemann@uke.de (J.H.); schwedhelm@uke.de (E.S.); l.schmidt-hutten@uke.de (L.S.-H.); h.stang@uke.de (H.S.); 2Institute DECIPHER, German-Chilean Institute for Research on Pulmonary Hypoxia and Its Health Sequelae, 20246 Hamburg, Germany; 3Primary Health Care of the Capital Area, 103 Reykjavik, Iceland; elinhelga@gmail.com; 4Faculty of Medicine, University of Iceland, 102 Reykjavik, Iceland; brynben@hi.is (B.B.); thorarig@landspitali.is (T.G.); 5National Heart and Lung Institute, Imperial College London, London SW3 6LY, UK; a.amaral@imperial.ac.uk; 6NIHR Imperial Biomedical Research Centre, London W2 1NY, UK; 7Sleep Department, Landspitali University Hospital of Iceland, 105 Reykjavik, Iceland; 8Unit for Nutrition Research, Landspitali University Hospital & Faculty of Food Science and Nutrition, University of Iceland, 102 Reykjavik, Iceland; ingigun@landspitali.is

**Keywords:** chronic obstructive lung disease, intermittent hypoxaemia, asymmetric dimethyl-arginine, ADMA, symmetric dimethylarginine, SDMA

## Abstract

Background: Chronic obstructive pulmonary disease (COPD) and obstructive sleep apnoea (OSA) are common chronic diseases that are associated with chronic and intermittent hypoxemia, respectively. Patients affected by the overlap of COPD and OSA have a particularly unfavourable prognosis. The L-arginine/nitric oxide (NO) pathway plays an important role in regulating pulmonary vascular function. Asymmetric (ADMA) and symmetric dimethylarginine (SDMA) interfere with NO production. Methods: We analysed the serum concentrations of ADMA, SDMA, L-arginine, L-citrulline, and L-ornithine in a large sample of the Icelandic general population together with chronic airflow obstruction (CAO), a key physiological marker of COPD that was assessed by post-bronchodilator spirometry (FEV1/FVC < LLN). OSA risk was determined by the multivariable apnoea prediction (MAP) index. Results: 713 individuals were analysed, of whom 78 (10.9%) showed CAO and 215 (30%) had MAP > 0.5. SDMA was significantly higher in individuals with CAO (0.518 [0.461–0.616] vs. 0.494 [0.441–0.565] µmol/L; *p* = 0.005), but ADMA was not. However, ADMA was significantly associated with decreasing FEV1 percent predicted among those with CAO (*p* = 0.002). ADMA was 0.50 (0.44–0.56) µmol/L in MAP ≤ 0.5 versus 0.52 (0.46–0.58) µmol/L in MAP > 0.5 (*p* = 0.008). SDMA was 0.49 (0.44–0.56) µmol/L versus 0.51 (0.46–0.60) µmol/L, respectively (*p* = 0.004). The highest values for ADMA and SDMA were observed in individuals with overlap of CAO and MAP > 0.5, which was accompanied by lower L-citrulline levels. Conclusions: The plasma concentrations of ADMA and SDMA are elevated in COPD patients with concomitant intermittent hypoxaemia. This may account for impaired pulmonary NO production, enhanced pulmonary vasoconstriction, and disease progression.

## 1. Introduction

Chronic obstructive pulmonary disease (COPD) and obstructive sleep apnoea (OSA) are both very common chronic diseases. The prevalence of COPD is at least 10% [1] and that of OSA is even higher [2], with nearly 425 million adults aged 30–69 years worldwide being estimated to have OSA based on an apnoea-hypopnoea index (AHI) of 15 or more events per hour [3]. Severe COPD is associated with chronic hypoxia [4], and OSA is characterized by intermittent hypoxaemia [5]. Patients with overlapping COPD and OSA show a significantly more rapid decline of lung function, more systemic cardiovascular comorbidity, and a higher mortality rate [6,7].

Hypoxia is known to act on the pulmonary vasculature causing vasoconstriction, which in turn induces cardiac hypertrophy and failure, and ensuing systemic co-morbidities [8]. We and others have shown that chronic and intermittent hypoxia leads to perturbations in the pulmonary L-arginine/dimethylarginine/nitric oxide (NO) pathway that result in elevated plasma concentrations of asymmetric dimethylarginine (ADMA), a major endogenous inhibitor of endothelial NO release [9]. Figure 1 shows a graphical representation of the pathway. 

ADMA directly interferes with NO synthesis by competitively displacing L-arginine from its substrate binding site, causing impaired endothelium-dependent vasodilation [10]. Both, symmetric dimethylarginine (SDMA) and ADMA inhibit cellular L-arginine uptake [11]. Besides generating NO, NO synthase (NOS) activity results in the conversion of L-arginine to L-citrulline. L-arginine is also a substrate for arginase, which converts it to L-ornithine [12]. Chronic and intermittent hypoxia leads to elevations of ADMA secondary to downregulation of its major metabolizing enzyme, dimethylarginine dimethylaminohydrolase (DDAH) [13]. Accordingly, genetic DDAH deletion leads to enhanced hypoxic vasoconstriction and pulmonary hypertension [14], whilst DDAH overexpression attenuates the sustained phase of hypoxic pulmonary vasoconstriction [15]. In humans, chronic intermittent hypoxia causes gradual elevation of ADMA over time. Individuals with high ADMA levels are at high risk of developing hypoxia-induced pulmonary arterial hypertension [16]. Common single nucleotide polymorphisms in the DDAH1 and DDAH2 genes predispose humans to high ADMA levels and hypoxia-induced pulmonary arterial hypertension [17].

Some small, cross-sectional studies have suggested that ADMA plasma concentration may be elevated in COPD and OSA. However, others have found no difference in ADMA levels between patients and controls (reviewed in detail in [9]). Very little information is available on the role of this biomolecule in people with overlapping COPD and OSA, with two small cross-sectional studies reporting inconclusive findings on ADMA plasma concentrations [18,19]. To investigate the role of ADMA in people with COPD and OSA, we measured serum concentrations of ADMA, along with SDMA, L-arginine and its metabolites, L-citrulline and L-ornithine, in a large population-based sample of Icelandic adults of the Burden of Obstructive Lung Disease (BOLD) study. 

## 2. Materials and Methods

### 2.1. Study Participants

BOLD is a multinational study set up to estimate the prevalence of chronic airflow obstruction, key characteristics of COPD, and identify its risk factors in general populations of adults aged ≥40 years around the world [20]. In Iceland, additional tools were used to screen for the presence of sleep-associated symptoms and estimate the risk of OSA [21]. After written informed consent was obtained, participants completed standardized questionnaires about their health, including sleep characteristics. The BOLD study in Iceland was approved by the National Bioethics Committee and the Data Protection Authority of Iceland (04-080). The present sub-study on biomarkers of the L-arginine/dimethylarginine/NO pathway received separate approval (FS-21-059-S1). 

### 2.2. Lung Function

Forced expiratory volume in one second (FEV1) and forced vital capacity (FVC) were measured using spirometry [20]. Pre- and post-bronchodilator testing was conducted with the participant in a sitting position wearing a nose clip and a disposable mouthpiece attached to the NDD Easy One™ spirometer (ndd Medizintechnik, Zurich, Switzerland). Chronic airflow obstruction (CAO) was defined as a post-bronchodilator FEV1/FVC ratio below the lower limit of normal (LLN), as described previously [22]. 

### 2.3. Sleep

Subjective sleep quality was assessed using the Basic Nordic Sleep Questionnaire (BNSQ) [23]. Daytime sleepiness was evaluated using the Epworth Sleepiness Scale (ESS) where >10 points were considered indicative of excessive sleepiness [24]. Additionally, as part of the BNSQ subjects responded to the statements “I feel sleepy during the day?” and “I feel rested during the day” on a 5-point scale from “never/almost never” to “every day/almost every day of the week”. Subjects were considered having significant sleepiness if they answered ≥3 times per week and not feeling rested if they answered <1 day per week, respectively. The probability of study participants suffering from obstructive sleep apnoea syndrome (OSA) was assessed by the Multivariable Apnoea Prediction (MAP) Index [25]. A cut-off value of 0.5 was used to define subjects at high risk of OSA as published before [21].

### 2.4. Potential Confounders

Doctor diagnoses of hypertension, cardiovascular disease and type 2 diabetes mellitus were self-reported. Smoking status was also self-reported. Current smokers were defined as those who had smoked more than 20 packs of cigarettes or more than one cigarette per day for at least one year and were smoking at the time of the interview. Those who had stopped smoking before the interview were considered former smokers. Never smokers were those who had never smoked or had smoked less than 20 packs of cigarettes in a lifetime.

### 2.5. Quantification of ADMA, SDMA, L-Arginine and Related Metabolites by LC-MS/MS

Blood was drawn from an antecubital vein at a similar time of the day. Samples were collected into SST serum tubes (Greiner, Kremsmuenster, Austria), allowed to stand for 20 min, and centrifuged for 10 min (3000 rpm). After separation, serum samples were stored at −20 °C until analysis.

Validated protocols for ultra-performance liquid chromatography-tandem mass spectrometry (UPLC-MS/MS) were used to quantify ADMA, SDMA, L-arginine, L-citrulline, and L-ornithine in serum. Briefly, 25 μL of serum were diluted in 100 µL methanol to which stable isotope labelled internal standards had been added. Subsequently, the compounds were converted into their butyl ester derivatives as described elsewhere [26]. Quantification of analytes was performed on a Waters UPLC-MS/MS platform (Xevo TQ-S cronos, Waters GmbH, Eschborn, Germany) applying an ACQUITY UPLC BEH C18 column (2.1 × 50 mm, 1.7 μm, Waters GmbH) for chromatographic separation. The coefficient of variation for the quality control samples was below 6% for all compounds.

Serum creatinine concentration was measured using a standard routine analysis method in the central clinical chemistry laboratory. Estimated glomerular filtration rate (eGFR) was calculated using the 2021 CKD-EPI formula [27]. 

### 2.6. Statistical Analyses

Categorical variables were summarized using frequencies and percentages and compared between groups using chi-squared tests. Continuous variables were summarized using means and standard deviations and compared between groups using Student’s *t* test. Biomarker levels were not normally distributed and therefore were summarized using medians and interquartile (IQR) range and compared between groups using a Kruskal–Wallis test. We assessed the association of CAO with each of the metabolites using logistic regression, first without adjustment (model 1) and then adjusting for age, sex, smoking status (model 2), body mass index (BMI) (model 3), and eGFR (model 4). Similar methods were used to explore the association between the biomarkers levels and high/low risk of OSA (MAP index). In an exploratory analysis, we assessed the association of the L-arginine-associated metabolites with CAO severity based on decreasing FEV1 percent predicted. A *p*-value < 0.05 was considered statistically significant. All statistics were calculated using Stata, version 16.0 (Stata Corporation, College Station, TX, USA).

## 3. Results

### 3.1. Participants’ Characteristics

A total of 758 participants had completed the study core questionnaire and had usable spirometry. Data for MAP index were missing for 16 BOLD participants, and bio-marker concentrations could not be obtained for 29 participants. Therefore, the current analyses are based on 713 participants (Figure 2). 

Study participants had a mean age of 56 ± 12 years; 47% were females. Table 1 displays the characteristics of the study population. There was a high prevalence of overweight and obesity, with an overall mean BMI of 28.0 ± 4.9 kg/m^2^. BMI was significantly higher in men than in women. About 60% of the study participants were current or former smokers. The prevalence of concomitant cardiometabolic diseases was 25% for hypertension, 15% for cardiovascular disease, and 3% for diabetes, with no significant differences between sexes. 

### 3.2. Prevalence of Chronic Airflow Obstruction, High OSA Risk, and Sleep-Related Symptoms

Appendix A gives an overview of variables relating to chronic airflow obstruction and OSA. Chronic airflow obstruction was present in 78 study participants (10.9%), with a higher prevalence in women than in men. Daytime sleepiness as assessed using the Epworth Sleepiness Score (ESS) was significantly higher in men than in women (6.4 ± 3.9 vs. 5.7 ± 3.9, *p* = 0.017). However, the proportion with ESS > 10 was 15.8% in the whole sample, with no significant sex difference. Restless legs syndrome was more frequently reported in women than in men (25% vs. 13%, *p* < 0.001). The overall mean MAP index was 0.36 ± 0.23, with 215 study participants (30%) having a MAP index > 0.5 (51% of the men and 7% of the women). Among the 78 study participants with CAO, 19 individuals (2.6%) also had MAP index > 0.5 and thus an overlap of both diseases (Figure 3). These 19 individuals had a mean BMI of almost 30, the majority had doctor-diagnosed hypertension and one third had concomitant cardiovascular disease (Table 2).

### 3.3. L-Arginine-Related Biomarkers: Baseline Serum Concentrations and Association with Concomitant Diseases

Table 1 reports the mean biomarker concentrations in this cohort. We noted a slight, but significant sex difference in SDMA and L-ornithine with higher values in men than in women. 

Except for L-arginine, all biomarkers showed increasing concentrations with age (Appendix A). SDMA and L-arginine were also associated with smoking status. The association between BMI and SDMA was J-shaped with the highest values among patients with BMI below 20.0 kg/m^2^. These 13 were older, and 6 (46%) of them had CAO (Appendix A). Study participants with hypertension had significantly higher ADMA, SDMA, and L-citrulline levels (Appendix A). The presence of cardiovascular disease was associated with SDMA and L-citrulline, and diabetes mellitus was associated with L-citrulline levels.

### 3.4. L-Arginine-Related Biomarkers in Relation to Chronic Airflow Obstruction (CAO)

There was a highly significant difference in SDMA serum concentration between participants with and without CAO (0.518 [0.461–0.616] vs. 0.494 [0.441–0.565] µmol/L; *p* = 0.005), but no significant difference in ADMA concentration (0.516 [0.447–0.594] vs. 0.505 [0.447–0.557] µmol/L; *p* = 0.20) (Table 3 and Figure 4a). All other biomarkers showed no significant differences (Table 3).

In univariable regression analysis, we found a significantly increased odds for CAO in study participants with serum SDMA in the highest quartile (OR = 2.78, 95% CI 1.43–5.37, *p* = 0.002). This association remained significant after adjusting for age, sex, and smoking status, but it lost significance when further adjusted for BMI or eGFR (Appendix A). No associations with CAO were observed for the other biomarkers. 

In an exploratory analysis addressing the association of biomarkers with decreasing FEV1 percent predicted among people with CAO (N = 78), ADMA serum concentration showed an incremental, highly significant increase with decreasing FEV1 percent predicted, whilst there was no significant trend for SDMA (Figure 4b,c) nor for the other biomarkers (Table 3). Seven study participants did not have FEV1 below 80% predicted when using a fixed cut-off in this analysis.

### 3.5. L-Arginine-Related Biomarkers in Relation to Sleep-Related Symptoms

None of the biomarkers showed a significant difference between study participants with ESS scores equal to or below 10 as compared to above 10 (Appendix A). ADMA was significantly higher in individuals who reported feeling sleepy more than three days per week (ADMA, 0.50 (0.44–0.55) vs. 0.53 (0.46–0.58) µmol/L; *p* < 0.03); for SDMA, a similar, non-significant trend was observed (0.49 (0.44–0.57) vs. 0.51 (0.46–0.59) µmol/L; *p* = 0.07). 

There was a significant association of ADMA and SDMA with MAP index (Figure 5). ADMA was 0.50 (0.44–0.56) µmol/L in individuals with MAP ≤0.5 versus 0.52 (0.46–0.58) µmol/L in those with MAP > 0.5 (*p* = 0.008). SDMA was 0.49 (0.44–0.56) µmol/L versus 0.51 (0.46–0.60) µmol/L in both groups, respectively (*p* = 0.004). For both biomarkers, the probability of having a MAP index ≤ 0.5 increased with increasing quartiles of biomarker serum levels; this association was retained after adjustment for eGFR (Appendix A). Except for L-citrulline, which showed a significantly increased odds for sleep apnoea in the highest quartile, no consistent trends over quartiles were observed for the other biomarkers.

### 3.6. L-Arginine-Related Biomarkers in the Overlap of Chronic Airway Obstruction with Sleep Apnoea

We observed significantly higher serum concentrations of ADMA and SDMA in the 19 patients with CAO and concomitant signs of OSA (i.e., MAP > 0.5). Both biomarkers were lowest in study participants with neither of the two health problems, slightly elevated in those with either CAO or sleep apnoea, and clearly elevated in the overlap group (Figure 6). There was also a significant trend for lower L-citrulline in the four groups, whilst none of the other biomarkers or ratios showed significant differences between the groups (Table 2).

## 4. Discussion

This study shows that the L-arginine/dimethylarginine/nitric oxide pathway is perturbed in chronic airflow obstruction and further so if the airway obstruction is severe. This association is strongest in the subgroup who in addition to CAO are likely to have OSA, i.e., intermittent hypoxaemia.

With a high number of participants and a population-based approach, our study provides strong evidence of the relationship between ADMA and SDMA concentrations and COPD. Previous studies haven generated inconclusive data on the potential role of L-arginine and its methylated derivatives in COPD. Ruzsics and co-workers reported significant elevations of ADMA, SDMA, and L-arginine, all measured by high performance liquid chromatography (HPLC), in a small group of 32 stable COPD patients who were diagnosed using the fixed cut-off of FEV1/FVC < LLN as compared to 30 healthy individuals [28]. In their study, all three amino acid metabolites were significantly further elevated in 12 patients with acute exacerbation of COPD versus those with stable COPD, and ADMA was inversely correlated with partial capillary oxygen pressure. Another study in 74 COPD patients revealed a positive correlation of ADMA with airway resistance assessed using FEV1/FVC < LLN [29]. In a cross-sectional comparison of 80 COPD patients diagnosed using a fixed cut-off of FEV1/FVC <0.70 with 40 healthy controls, Telo and co-workers [30] found no significant difference in ADMA plasma concentration. However, a subgroup of COPD patients with pulmonary arterial hypertension showed elevated ADMA levels. No significant difference in ADMA between COPD patients (using FEV1/FVC < LLN) and controls was reported by Costanzo and co-workers [31]. We have recently reviewed these discrepant findings from clinical studies [9]. In general, previous studies were not based on general population samples, and not in all studies was post-bronchodilator spirometry used to assess CAO. In addition, a fixed cut-off for FEV1/FVC <0.70 was used in some studies, whilst others applied a ratio of FEV1/FVC < LLN, which results in fewer cases of CAO in elderly individuals and has been recommended for screening purposes [22]. Moreover, concomitant diseases, not the least of it being sleep apnoea, have not been documented in most of the previous studies, and, finally, the quantification of ADMA often relied on HPLC rather than the analytical gold standard of LC-MS/MS that was applied in the present study.

Similarly, previous studies in sleep apnoea syndrome generated conflicting results. İn and co-workers [32] reported higher ADMA but lower L-arginine serum levels in OSA patients than in healthy controls. These differences were unrelated to traditional cardiovascular risk factors, as OSA patients with or without risk factors showed similar differences to controls. Others found discrepant results (reviewed by [9]), which may relate to factors such as concomitant cardiovascular diseases and obesity, which have been shown to affect ADMA levels [33,34]. In addition, none of the studies cited above reported co-morbidity rates of COPD and OSA.

In the present study we observed a significant association of SDMA, but not ADMA to BMI. This association was J-shaped with highest SDMA concentrations in individuals with lowest BMI. Although dimethylarginines have often been found to be elevated in obese people, malnourishment may be associated with higher protein L-arginine dimethylation and, thus, higher rates of generation of both, ADMA and SDMA [35]. This may be of importance for our present study, since malnutrition has been shown to be an independent predictor of lung function in hospitalized COPD patients [36]. In addition, we found both dimethylarginines to be associated with hypertension in the present study.

The highest ADMA and SDMA levels were present in the 19 study participants with concomitant signs of both CAO and OSA. Their high ADMA and SDMA concentrations and low L-citrulline levels may well be indicative of low NO synthase activity. Larger studies of subjects with the overlap syndrome are needed to confirm if elevated concentrations of ADMA and SDMA may account for impaired pulmonary NO production and the unfavourable disease progression seen in the overlap syndrome if untreated [7].

Both ADMA and SDMA mechanistically contribute to low NO generation in the hypoxic lungs, which in turn has been shown to contribute to inflammatory mechanisms and lung remodelling [37]. In addition, both molecules are prospective markers of mortality risk in the general population [38,39]. Therefore, whilst not being useful as disease-specific diagnostic markers, elevated ADMA and SDMA may not only point to pathophysiologically important changes of biochemical pathways but may also relate to long-term risk determination in chronic lung diseases associated with intermittent hypoxia. Indeed, using a prospective study design with six years of follow-up, Vögeli and co-workers [40] observed a significant association of ADMA with long-term mortality of patients initially included with acutely exacerbated COPD. The association of ADMA with mortality was retained in their study after multivariable adjustment of the regression, whilst SDMA was only associated with mortality in univariate models, and L-arginine showed no significant association.

The mean serum concentration of L-arginine was higher than previously published reference ranges [41]. This difference may be due to high contribution of dietary proteins to total energy intake as well as high intake of fish and seafood in the Icelandic population [42]. L arginine content in dietary protein from different sources varies from 3 to 15%. Soy protein, nuts, fish and seafood are comparatively rich in L-arginine [43]. By contrast, cereals are relatively poor sources of L-arginine, with only 3–4% of their total protein being L-arginine. All other metabolites were within published reference ranges. Based on high L-arginine levels, the L arginine/ADMA ratio was also higher than previously reported values in the healthy American population [41].

In our study the probability of OSA was assessed by calculating the MAP index, which has been reported to be a reliable estimate of OSA risk; however, definite diagnosis would require overnight polysomnographic studies which were not performed in this study. It would therefore be interesting to determine ADMA and SDMA concentrations in a large sample of people with OSA confirmed with overnight sleep studies, and to assess the effect of PAP treatment on these biomarker levels. Further, we used self-reported cardiovascular disease, hypertension, and type 2 diabetes mellitus as concomitant diagnoses, which might be strengthened by doctor-confirmed diagnoses in a future study. In addition, analysing the prospective roles of ADMA and SDMA for outcome of COPD in a large, prospective study cohort would help to better understand their roles as predictive markers of mortality and, potentially, lung function decline over time.

## 5. Conclusions

Our data show that the L-arginine/dimethylarginine/NO pathway is dysregulated in chronic respiratory diseases in a manner relating to the presence of intermittent hypoxaemia. Elevated concentrations of ADMA and SDMA may account for impaired pulmonary NO production, driving pulmonary vasoconstriction and disease progression. In addition, elevated systemic serum concentrations of these markers may account for systemic cardiovascular disease and, thereby, be linked to poor prognosis. Future, prospective studies will be needed to prove this hypothesis.

## Figures and Tables

**Figure 1 jcm-12-05230-f001:**
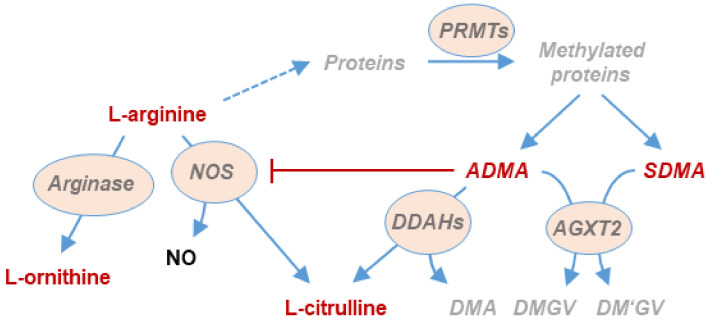
Schematic representation of the L-arginine/dimethylarginine/nitric oxide pathway. L-arginine is metabolized to nitric oxide (NO) and L-citrulline by NO synthases (NOS) or to L-ornithine and urea by arginases. In addition, L-arginine residues within specific proteins are subject to methylation by protein arginine N-methyltransferases (PRMT’s). When dimethylated proteins are degraded, asymmetric (ADMA) and symmetric dimethylarginine (SDMA) are released. ADMA is a competitive inhibitor of nitric oxide synthases (NOS). ADMA, but not SDMA, is degraded by dimethylarginine dimethylaminohydrolases (DDAH) to L-citrulline and dimethylamine (DMA). Both dimethylarginines may also be cleaved by alanine glyoxylate aminotransferase 2 (AGXT2), resulting in the formation of symmetric or asymmetric dimethylguanidinovaleric acid (DMGV and DM’GV).

**Figure 2 jcm-12-05230-f002:**
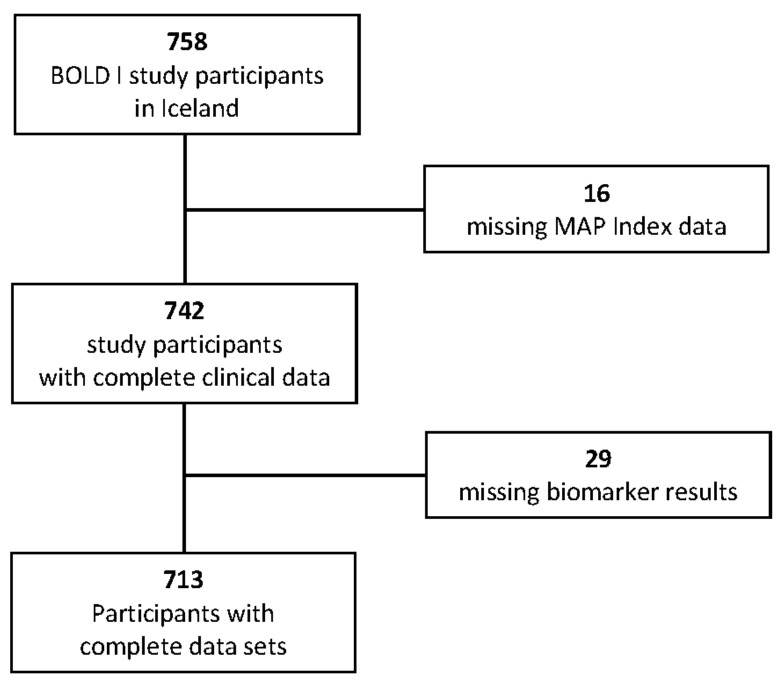
CONSORT diagram depicting the study flow.

**Figure 3 jcm-12-05230-f003:**
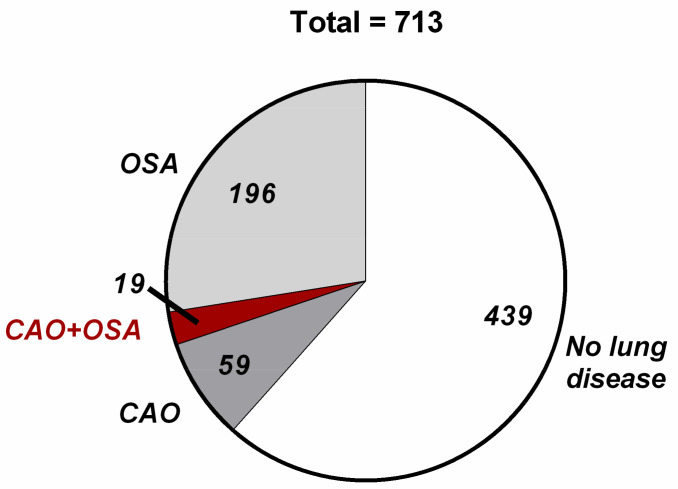
Proportions of study participants with chronic airway obstruction (CAO), symptoms of obstructive sleep apnoea (OSA), or both (CAO + OSA), or no lung disease. Numbers are absolute numbers of study participants in each category.

**Figure 4 jcm-12-05230-f004:**
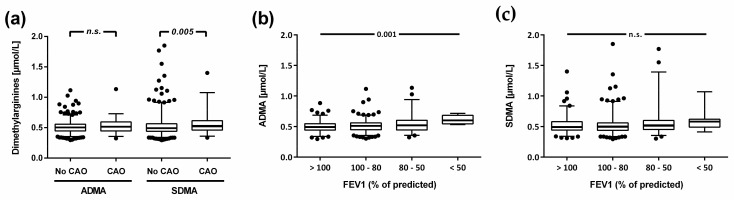
Serum concentrations of ADMA and SDMA in study participants with and without chronic airway obstruction (CAO), respectively (**a**), and in subgroups of people with CAO in relation to levels of FEV1 percent predicted (**b**,**c**). Data show median and interquartile range, with vertical whiskers indicating the 2.5th and 97.5th percentiles. Extreme values are plotted individually.

**Figure 5 jcm-12-05230-f005:**
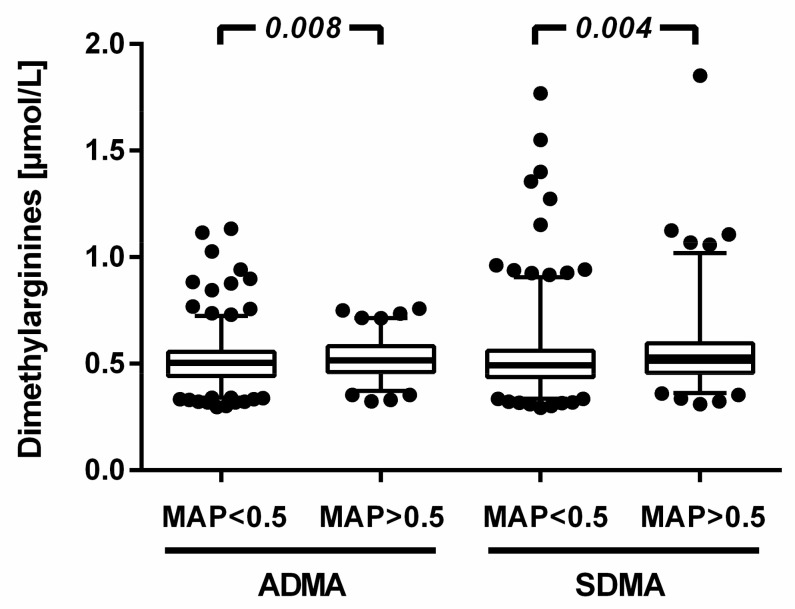
Serum concentrations of ADMA and SDMA in study participants with MAP ≤ 0.5 and >0.5, respectively. Data show median and interquartile range, with vertical whiskers indicating the 2.5th and 97.5th percentiles. Extreme values are plotted individually.

**Figure 6 jcm-12-05230-f006:**
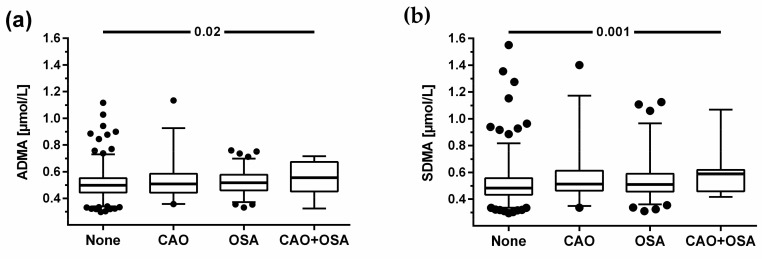
Serum concentrations of ADMA (**a**) and SDMA (**b**) in study participants with chronic airflow obstruction (CAO), obstructive sleep apnoea (OSA), or both (CAO + OSA), or without lung disease. Data show median and interquartile range, with vertical whiskers indicating the 2.5th and 97.5th percentiles. Extreme values are plotted individually.

**Table 1 jcm-12-05230-t001:** Characteristics of the study population by sex.

Variable	Total	Female	Male	*p* *
N (%)	713 (100)	336 (47.1)	377 (52.9)	n.a.
Demographic parameters				
Age (years)	56 ± 12	57 ± 12	56 ± 12	0.36
BMI (kg/m^2^)	28.0 ± 4.9	27.6 ± 5.4	28.3 ± 4.4	**0.04**
BMI <25 kg/m^2^	201 (28.2)	122 (36.3)	79 (20.9)	**<0.001**
BMI 25–29.9 kg/m^2^	312 (43.8)	124 (36.9)	188 (49.9)	
BMI ≥30 kg/m^2^	200 (28.0)	90 (26.8)	110 (29.2)	
Smoking status				
Never	278 (39.1)	132 (39.3)	146 (38.9)	0.100
Former	302 (42.5)	132 (39.3)	170 (45.3)	
Current	131 (18.4)	72 (21.4)	59 (15.7)	
Pack-years, current smokers	26.0 ± 16.7	22.4 ± 13.4	30.3 ± 19.1	**0.006**
Pack-years, ever smokers	9.9 ± 26.1	6.6 ± 11.4	12.7 ± 33.6	**0.005**
Concomitant cardiometabolic diseases				
Hypertension	177 (25.3)	95 (28.6)	82 (22.2)	0.052
CVD	106 (14.9)	48 (14.3)	58 (15.4)	0.670
Diabetes	20 (2.8)	10 (3.0)	10 (2.7)	0.970
Biomarker levels				
ADMA (µmol/L)	0.51 [0.45, 0.56]	0.51 [0.45, 0.55]	0.51 [0.44, 0.56]	0.920
SDMA (µmol/L)	0.50 [0.44, 0.57]	0.48 [0.42, 0.56]	0.51 [0.45, 0.58]	**0.001**
L-Arginine (µmol/L)	144.0 [126.9, 163.2]	142.8 [125.8, 162.5]	145.0 [128.9, 163.6]	0.870
L-Citrulline (µmol/L)	39.5 [33.1, 47.3]	39.0 [32.4, 47.6]	39.8 [33.7, 47.3]	0.460
L-Ornithine (µmol/L)	80.4 [67.0, 95.3]	77.4 [65.1, 94.5]	82.2 [68.3, 95.5]	**0.040**
Arg/ADMA Ratio	288.7 [251.2, 330.1]	288.2 [253.2, 327.2]	289.7 [249.5, 333.4]	0.880
Cit/Arg Ratio	0.27 [0.23, 0.33]	0.27 [0.23, 0.33]	0.28 [0.23, 0.33]	0.450
Orn/Arg Ratio	0.55 [0.46, 0.67]	0.54 [0.45, 0.65]	0.56 [0.47, 0.68]	**0.027**

Data are presented as mean ± standard deviation and median with interquartile ranges in square brackets for continuous variables as appropriate. Categorical variables are presented as number of observations (N) with the percent value given in brackets. * *p* values from Student’s *t* test for means, Kruskal–Wallis test for medians (continuous), and chi-squared test (categorical) comparing males with females. Abbreviations: ADMA, asymmetric dimethylarginine; Arg, L-arginine; BMI, body mass index; Cit, L-citrulline; CVD, cardiovascular disease; Orn, L-ornithine; SDMA, symmetric dimethylarginine.

**Table 2 jcm-12-05230-t002:** General characteristics, smoking status, health, and biomarker concentrations in study participants with or without chronic airflow obstruction (CAO), high obstructive sleep apnoea risk (MAP), or both.

	−MAP −CAO	−MAP +CAO	+MAP −CAO	+MAP +CAO	*p* *
Demographics and anthropometrics					
N (% males)	439 (39.2)	59 (23.7)	196 (88.3)	19 (94.7)	-
Age (years)	54 ± 11.0	62.3 ± 12.7	60.0 ± 11.7	65.9 ± 10.5	**<0.001**
BMI (kg/m^2^)	26.6 ± 3.7	24.8 ± 4.0	31.7 ± 5.4	29.7 ± 4.3	**<0.001**
BMI < 25 kg/m^2^	154 (35.1)	34 (57.6)	11 (5.6)	2 (10.5)	**<0.001**
BMI 25–29.9 kg/m^2^	213 (48.5)	19 (32.2)	70 (35.7)	10 (52.6)	
BMI ≥ 30 kg/m^2^	72 (16.4)	6 (10.2)	115 (58.7)	7 (36.8)	
Smoking status					
Never	186 (42.4)	16 (27.1)	69 (35.6)	7 (36.8)	0.112
Former	174 (39.6)	26 (44.0)	93 (47.9)	9 (47.4)	
Current	79 (18.0)	17 (28.8)	32 (16.5)	3 (15.8)	
Pack-years, current smokers	22.0 ± 15.6	33.6 ± 11.4	30.6 ± 19.4	36.0 ± 11.4	**0.008**
Pack-years, ever smokers	14.2 ± 21.9	26.1 ± 18.5	24.0 ± 50.3	43.9 ± 31.3	**0.008**
Concomitant cardiovascular and metabolic diseases					
Hypertension (%)	92 (21.3)	19 (32.2)	56 (29.2)	10 (55.6)	**0.001**
Cardiovascular disease (%)	45 (10.3)	15 (25.4)	40 (20.5)	6 (31.6)	**<0.001**
Diabetes (%)	8 (1.8)	2 (3.4)	9 (4.6)	1 (5.6)	0.226
Biomarkers					
ADMA	0.5 (0.44, 0.55)	0.50 (0.44, 0.58)	0.52 (0.46, 0.58)	0.55 (0.45, 0.67)	**0.019**
SDMA	0.48 (0.43, 0.56)	0.51 (0.46, 0.61)	0.51 (0.46, 0.59)	0.58 (0.46, 0.62)	**0.0007**
L-Arginine	142.5 (126.4, 161.6)	146.1 (125.1, 172.9)	148.5 (129.4, 165.1)	138.1 (130.2, 164.0)	0.34
L-Citrulline	38.2 (32.4, 46.3)	41.4 (33.5, 52.5)	40.9 (33.9, 49.5)	37.8 (32.7, 50.2)	**0.028**
L-Ornithine	78.7 (66.0, 93.5)	79.4 (62.8, 101.2)	84.5 (70.3, 96.0)	81.4 (69.2, 104.5)	0.07
Arg/ADMA Ratio	287.8 (253.8, 327.4)	289.7 (247.0, 342.7)	294.1 (247.3, 335.0)	259.8 (212.1, 323.6)	0.40
Cit/Arg Ratio	0.27 (0.23, 0.33)	0.28 (0.24, 0.35)	0.28 (0.22, 0.34)	0.27 (0.21, 0.32)	0.50
Orn/Arg Ratio	0.55 (0.45, 0.67)	0.54 (0.47, 0.65)	0.56 (0.47, 0.66)	0.57 (0.50, 0.72)	0.39

* *p*-values derived from Kruskal Wallis test.

**Table 3 jcm-12-05230-t003:** Biomarker concentrations in the 78 study participants with chronic airflow obstruction (CAO) based on LLN compared to those without CAO and further divided based on % FEV1 predicted.

Biomarker/ Ratio	Chronic Airflow Obstruction (CAO)			Predicted FEV1 (% Based on NHANES III Reference Values) among the 78 Participants with Chronic Airflow Obstruction				
	Yes	No	*p* *	>100	80–100	50–80	<50	*p* *
N	78	635	n.a.	7	25	33	13	n.a.
ADMA	0.52 (0.45–0.59)	0.51 (0.45–0.56)	0.19	0.52 (0.45, 0.58)	0.48 (0.43, 0.52)	0.51 (0.43, 0.60)	0.60 (0.55, 0.67)	**0.001**
SDMA	0.52 (0.46–0.62)	0.49 (0.44–0.56)	**0.005**	0.62 (0.50, 0.81)	0.50 (0.45, 0.60)	0.52 (0.46, 0.60)	0.58 (0.50, 0.62)	0.204
L-Arginine	145.3 (125.7–167.1)	143.8 (127.4–163.1)	0.77	130.9 (116.2, 146.1)	138.1 (121.4, 159.7)	147.8 (126.3, 172.9)	158.0 (138.6, 178.2)	0.305
L-Citrulline	41.0 (33.5–50.2)	39.3 (32.9–47.2)	0.18	52.5 (46.4, 66.1)	40.1 (37.4, 46.3)	40.2 (32.2, 47.7)	42.0 (32.9, 51.2)	0.102
L-Ornithine	79.7 (65.3–101.2)	80.4 (67.1–95.1)	0.75	88.0 (66.8, 89.9)	79.1 (61.7, 93.6)	80.8 (65.7, 106.7)	79.4 (75.0, 93.4)	0.870
Arg/ADMA Ratio	283.9 (244.2–337.1)	289.0 (252.6–329.6)	0.49	287.1 (203.1, 314.3)	314.6 (258.0, 342.7)	295.9 (247.0, 339.4)	255.6 (222.2, 273.7)	0.222
Cit/Arg Ratio	0.28 (0.24–0.35)	0.27 (0.23–0.33)	0.32	0.44 (0.36, 0.54)	0.28 (0.24, 0.32)	0.28 (0.24, 0.32)	0.26 (0.21, 0.32)	**0.016**
Orn/Arg Ratio	0.57 (0.48–0.66)	0.55 (0.46–0.67)	0.74	0.56 (0.51, 0.68)	0.56 (0.49, 0.64)	0.57 (0.47, 0.67)	0.51 (0.43, 0.63)	0.570

Data are presented as median with interquartile range in brackets. Biomarker concentrations are in µmol/L. * *p*-values are from Kruskal–Wallis test. Abbreviations: ADMA, asymmetric dimethylarginine; SDMA, symmetric dimethylarginine; Arg/ADMA ratio, ratio of serum L-arginine over ADMA concentrations; Cit/Arg ratio, ratio of serum L-citrulline over L-arginine concentrations; NHANES, National Health and Nutrition Examination Survey; Orn/Arg ratio, ratio of serum L-ornithine over L-arginine concentrations; MAP, multivariable apnoea prediction index; CAO, chronic airflow obstruction as defined by FEV1/FVC < LLN.

## Data Availability

The data generated in this study are available in this article and in the Appendix A provided.

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
