# Peer review of "Biomarkers of the L-Arginine/Dimethylarginine/Nitric Oxide Pathway in People with Chronic Airflow Obstruction and Obstructive Sleep Apnoea"

_jcm, 2023, doi:10.3390/jcm12165230_

Round 1

Reviewer 1 Report

This is a well-written manuscript and a thoroughly researched subject matter.  The authors successfully describe a correlation between OSA-CAO and SDMA and ADMA levels, thereby identifying a potentially impactful biomarker series for future patients. 

Some minor comments/suggestions:

Any comment on the gender difference found for L-ornithine and SDMA?

To enhance translatability, do the authors have any comment on the study’s prevalence of OSA, daytime sleepiness, and restless leg syndrome in comparison to known prevalence in the general public?

Page 6, there is a typographical error: CA0 (zero) versus CAO: “These 13 were older and 6 (46 %) of them had CA0 (Table S3).”

I would encourage the authors to highlight a study weakness of using self-reported CV, HTN, and DM2.  That said, the authors remark on the optimal (and next step) study using overnight polysomnography. 

Perhaps I missed it, but the authors state that some of their OSA subjects had ‘concomitant hypoxia’ but it is unclear how this was determined.

Lastly, it may be helpful to the reader if the authors would be allowed to include a figure that captures the NO, arginine, ADMA, and SDMA relationships.  

Author Response

Reviewer 1

This is a well-written manuscript and a thoroughly researched subject matter.  The authors successfully describe a correlation between OSA-CAO and SDMA and ADMA levels, thereby identifying a potentially impactful biomarker series for future patients. 

 Some minor comments/suggestions:

  1. Any comment on the gender difference found for L-ornithine and SDMA?

Reply: We have previously published plasma concentrations of ADMA, SDMA, and the L-arginine/ADMA ratio from the large, prospective community-based Framingham Offspring cohort. We found slightly different normal ranges for men and women and an age-dependent increase in concentrations. Sex differences in ornithine levels are less well described; however, in the present sample from the BOLD cohort, we report only minimal differences between men and women in SDMA, L-arnithine, and in ornithine/arginine ratio. In the context of the main focus of our present study, we decided to not elaborate on this sex difference in detail.

  1. To enhance translatability, do the authors have any comment on the study’s prevalence of OSA, daytime sleepiness, and restless leg syndrome in comparison to known prevalence in the general public?

Reply: Our analyses in the Icelandic BOLD I cohort have been made in a population-based sample, as described in the Methods section of our manuscript. With prevalences of 10.9 % for CAO and 30.2 % for OSA in our study cohort, the prevalences of both conditions are in the range of those previously reported. It needs to be mentioned, though, that the most reliable data on CAO prevalences worldwide come from the BOLD study (Reference 1 of our manuscript), of which we investigated one national subgroup here. Further, the largest cross-sectional survey on OSA comes from one of our co-authors (TG). In summary, these data reassure us that our data may be translated in broader context to the general population. We have noted in the manuscript that BOLD is a population-based study (section on „Study Participants“).

  1. Page 6, there is a typographical error: CA0 (zero) versus CAO: “These 13 were older and 6 (46 %) of them had CA0 (Table S3).”

Reply: We thank the reviewer for carefully reading our manuscript and hinting to this typo. We have corrected it in the revised manuscript version.

  1. I would encourage the authors to highlight a study weakness of using self-reported CV, HTN, and DM2. That said, the authors remark on the optimal (and next step) study using overnight polysomnography. 

Reply: We thank the reviewer for this suggestion. Indeed, the BOLD Study being a multi-national epidemiological survey with strict rules for data curation set into force that applied to all study centers worldwide, CV, hypertension, and type 2 diabetes are self-reported diagnoses. We have extended the limitations section mentioned by the reviewer to inclulde the aspect that doctor diagnoses of concomitant diseases would strengthen the data of any future study on this topic.

  1. Perhaps I missed it, but the authors state that some of their OSA subjects had ‘concomitant hypoxia’ but it is unclear how this was determined.

Reply: We assume that the reviewer refers here tot he first paragraph of the Discussion section where we state: „This association is strongest in the subgroup who in addition to CAO are likely to have OSA with intermittent hypoxaemia.“ We intended to say that the concomitant presence of CAO and OSA is a sign that intermittent hypoxaemia is present. To make this clearer, we re-worded the sentence as follows: „This association is strongest in the subgroup who in addition to CAO are likely to have OSA, i.e., intermittent hypoxaemia.“

  1. Lastly, it may be helpful to the reader if the authors would be allowed to include a figure that captures the NO, arginine, ADMA, and SDMA relationships.

Reply: We appreciate this suggestion by the reviewer, and we acknowledge that the pathways analysed in our study are complex to readers who are not a priori familiar with L-arginine metabolism. We have therefore decided to take up the reviewer’s suggestion and included a graphical representation of L-arginine metabolic pathways in which we highlighted the metabolites measured in this study (new Figure 1).

Reviewer 2 Report

In the present study, the authors analysed the serum concentrations of ADMA, SDMA, L- arginine, L-citrulline, and L-ornithine in a large sample of the Icelandic general population together with chronic airflow obstruction CAO. OSA risk was also determined by the multivariable apnoea prediction (MAP) index. They found that SDMA was significantly higher in individuals with CAO. ADMA was significantly associated with decreasing FEV1 percent predicted among those with CAO. The highest values for ADMA and SDMA were observed in individuals with overlap of CAO and MAP > 0.5, which was accompanied by lower L-citrulline levels. They concluded that the plasma concentrations of ADMA and SDMA are elevated in COPD patients with concomitant intermittent hypoxaemia.This study is well organized, is a technical tour-de-force, and the results conclusive.

Author Response

Reviewer 2

In the present study, the authors analysed the serum concentrations of ADMA, SDMA, L- arginine, L-citrulline, and L-ornithine in a large sample of the Icelandic general population together with chronic airflow obstruction CAO. OSA risk was also determined by the multivariable apnoea prediction (MAP) index. They found that SDMA was significantly higher in individuals with CAO. ADMA was significantly associated with decreasing FEV1 percent predicted among those with CAO. The highest values for ADMA and SDMA were observed in individuals with overlap of CAO and MAP > 0.5, which was accompanied by lower L-citrulline levels. They concluded that the plasma concentrations of ADMA and SDMA are elevated in COPD patients with concomitant intermittent hypoxaemia. This study is well organized, is a technical tour-de-force, and the results conclusive.

Reply: We greatly appreciate the favourable comment of this reviewer on our manuscript.

Reviewer 3 Report

The study is well-designed with the predetermined outcomes.  Even though authors claim that the levels of ADMA and SDMA are different between healthy and disease groups, the difference is too small and the ranges overlap.  The diet significantly contributes to levels of L-arginine, ADMA and  SDMA in the blood.  Should stratifying the levels of L-arginine and ADMA and SDMA with diet if possible.

Author Response

Reviewer 3

The study is well-designed with the predetermined outcomes. Even though authors claim that the levels of ADMA and SDMA are different between healthy and disease groups, the difference is too small and the ranges overlap. The diet significantly contributes to levels of L-arginine, ADMA and  SDMA in the blood.  Should stratifying the levels of L-arginine and ADMA and SDMA with diet if possible.

Reply: We greatly appreciate the favourable comment of this reviewer on pour manuscript. We further agree that the absolute differences in ADMA and SDMA serum concentrations between patients and controls are not large. However, we would like to point to a previous, large, population-based cohort win which we showed that a 0.134 µmol/L difference in ADMA concentration is associated with an increase in risk of total mortality that equals that of a 2.4-year increase in age. Although it has not been definitely resolved why such small absolute differences in biomarker concentrations relate to such large increases in risk, this fact underlines the potential clinical importance of small differences in these biomarkers’ concentrations for individuals with chronic lung disease and concomitant intermittent hypoxaemia.